# Fine Structure and Optical Features of the Compound Eyes of Adult Female *Ceratosolen gravelyi* (Hymenoptera: Agaonidae)

**DOI:** 10.3390/insects16070682

**Published:** 2025-06-30

**Authors:** Hua Xie, Yan Shi, Shouxian Zhang, Yonghui Zhu, Subo Shao, Yuan Zhang, Pei Yang, Zongbo Li

**Affiliations:** 1Key Laboratory of Forest Disaster Warning and Control in Yunnan Province, Southwest Forestry University, Kunming 650224, China; hxie21527@outlook.com (H.X.); zsxdy@outlook.com (S.Z.); zyhsun11@outlook.com (Y.Z.); shaosubo1122@outlook.com (S.S.); yuanzhang@swfu.edu.cn (Y.Z.); 2Institute of Management Cadres, National Academy of Forestry and Grassland Administration, Beijing 102600, China; mdfshiyan@sohu.com; 3College of Traditional Chinese Medicine, Yunnan University of Chinese Medicine, Kunming 650500, China

**Keywords:** Agaonidae, vision, ultrastructure, TEM, apposition eye, fused rhabdom

## Abstract

Fig wasps and fig trees depend on each other to survive, and wasps locate their host trees with their sense of smell. However, the role that vision plays in locating host trees remains unclear. Here, we examined the eye structure of female *Ceratosolen gravelyi* wasps (which only pollinate *Ficus semicordata*) using scanning/transmission electron microscopy. Their oval-shaped eyes have 228–263 tiny eye units (ommatidia). Each unit has a clear lens, a four-part light-focusing structure, and light-sensing cells wrapped in pigment cells. The retinula cells form a rhabdom with nine cells per unit. Eight photoreceptors (R1–R8) form the rhabdom from the cone base to the basal matrix, and a ninth cell replaces R8 in the apical third of the rhabdom. Various optical measurements revealed that their eyes collect light efficiently. These features are consistent with their daytime activity, suggesting that their eyes process visual cues when approaching figs. This indicates that vision likely works with smell and touch to guide wasps to the right trees. Our findings thus enhance our understanding of how senses work together to mediate fig-wasp interactions.

## 1. Introduction

Insect compound eyes, which comprise numerous repeating units called ommatidia, exhibit marked structural and functional diversity across species [1,2,3]. The number of ommatidia varies drastically among species, from 29,247 in Odonata [4] and 16,000 in carpenter bees [5] to 800 in *Drosophila* [6] and a few dozen in miniature feather-winged beetles [7] and parasitoid wasps [8,9]. In extreme cases, such as parasitic lice (Phthiraptera), fleas (Siphonaptera), female scale insects (Hemiptera), and permanently subterranean species (e.g., termites, cave crickets, and beetles), eyes are severely reduced or absent [1,10]. Variation in eye morphology is observed both among closely related species (e.g., wider, more numerous ommatidia and larger eyes in *Drosophila mauritiana* subgroup species compared to *D. simulans* [11]) and within individual eyes (e.g., the dorsal rim, transitional, and regular ommatidia zones in *Megaphragma viggianii* [9]). Each ommatidium comprises a corneal lens, crystalline cone, light-sensitive rhabdom, and pigment cells, which work together to focus light, convert it into neural signals, and prevent optical crosstalk. Ommatidial elements further vary between species; facet diameters range from 180 µm (*Titanus giganteus*) [12] to 5.8 µm (*Trichogramma evanescens*) [13], rhabdomere lengths range from 400 µm (*Apis* drones) [14] to 24.29 µm (*T. evanescens*) [13], and pigment granule shapes range from spherical (normal) to ellipsoidal (miniature) [8,9,15]. This variation regulates light intake of the ommatidia, which directly affects both the spatial resolution and light sensitivity of the compound eye’s mosaic vision [16,17]. Compound eyes can be classified into three major types according to their design, apposition, optical superposition, and neural superposition eyes [1], which are each adapted to specific ecological needs; these different types of compound eyes provide insects with a broad visual field and acute motion detection and allow them to perform their activities under diverse light conditions [16,18].

In Hymenoptera, the apposition eye, characterized by the absence of a clear zone between the facet lens and retina, operates as an independent optical unit, detecting only light that enters through its lens. This structure is typical of diurnal species, including most wasps, bees, and ants [3,6,14,18,19,20,21]. However, some nocturnal Hymenoptera, such as the sweat bee *Megalopta genalis* (Halictidae) [22] and formicine ants *Myrmecia* spp. (Formicidae) [20], have retained this eye type despite their nocturnal activity. This contrasts with nocturnal moths, which have evolved superposition or neural superposition eyes for enhanced photon capture and improved sensitivity in low-light conditions [23,24]. Parasitoid chalcid wasps (superfamily Chalcidoidea) are hymenopterans whose larvae develop as parasitoids, feeding on or inside hosts (e.g., plants, insects, spiders), ultimately killing them; adults are free-living and serve key ecological roles as natural enemies [25]. Despite their significance, the compound eye structure has been characterized in only three species: *Trichogramma evanescens* [13,26], *Anaphes flavipes* [15], and *Megaphragma viggianii* [8,9,15]. Comparative analysis reveals significant structural variations across these lineages, including eye size, ommatidial density, regional specialization, and internal anatomy [8,9,13,15,26]. For example, two *Trichogramma* species exhibit reduced interommatidial angles, widening their visual field to enhance host egg location [13,15]. Such variations reflect evolutionary adaptations to species-specific lifestyles, behavioral ecology, and optical environmental constraints [3,27].

Pollinating fig wasps (Agaonidae), which are minute chalcid wasps, engage in obligate mutualisms with *Ficus* species (Moraceae), and each host fig relies on one, or rather often, several congeneric agaonid wasp species for pollination [28,29,30,31]. This symbiosis centers on the syconia (figs), an urn-shaped enclosed inflorescence progressing through five developmental stages: (A) pre-female, (B) receptive, (C) interfloral, (D) male, and (E) postfloral. The ostiole, a bract-sealed entry portal at the syconium apex, opens exclusively during stage B to admit mated female wasps carrying pollen. These wasps simultaneously pollinate female flowers and oviposit in select ovaries of the female flowers, where their larvae develop obligately. After some weeks, their offspring become adults. Winged females possess fully developed antennae and compound eyes and are capable of dispersing to new host figs, while flightless males retain vestigial sensory organs and are confined to natal syconia over their lifetime [29,32]. Pollinating wasps were mainly thought to be attracted specifically by species-specific volatile cues from stage B figs [32,33,34,35,36]. However, the surfaces of syconia at different developmental stages often display distinct characteristic color patterns and changes. For example, *Ficus semicordata* syconia transition through green, purple, and red hues (Figure 1). These visual cues may allow the assessment of syconia suitability for wasps. Crucially, mature, mated female wasps within ripe syconia likely rely on their compound eyes to detect light emitted through exit holes created by male fig wasps. Supporting the importance of vision, our empirical observations have revealed distinct phototactic behaviors, including positive phototaxis towards ambient light sources (e.g., window-proximal regions), chromatic discrimination capacity among colored traps (yellow > green > white), and stage-specific visual preference for daytime oviposition (Stage B > A/C/D syconia). More importantly, genomic analysis of two *Ceratosolen* species (*C. solmsi* and *C. marchali*) [37,38] has shown that their trichromatic vision is mediated by three photoreceptors peaking at 340 nm (UV), 430 nm (blue), and 535 nm (green), which is consistent with ancestral hymenopteran spectral sensitivity patterns [39]. These findings indicate that the validity of chemo-centric models of fig–wasp communication should be re-evaluated and that the role of vision in the fig–wasp mutualism might be underappreciated, especially for short-range discrimination in heterogeneous microhabitats [40]. Studies of the structural specialization of compound eyes, including ommatidial densities, optimized interommatidial angles, and rhabdom dimensions, are needed to clarify multimodal signaling integration (e.g., chemical, visual, tactile) in this conserved mutualism. Yet, no such studies have been conducted on fig wasp compound eyes.

Here, we characterize for the first time the ocular morphology and ultrastructural adaptations of adult female fig wasps *Ceratosolen gravelyi* using scanning electron microscopy (SEM) and transmission electron microscopy (TEM). As the sole pollinator of functionally dioecious *F. semicordata*, *C. gravelyi* locates stage B syconia via the species-specific attractant 4-methylanisole [41], a chemical signature distinct from those of other chalcid wasps and hymenopterans [35,36,42]. It performs dual biological roles: pollinating female syconia while simultaneously ovipositing and inducing galls in male syconia. Through anatomical measurements, we quantitatively analyzed key optical parameters and performed comparative optometric assessments with three-minute parasitoids: *T. evanescens* [13], *A. flavipes* [15], and *M. viggianii* [9,15]. Finally, we examine the functional implications of small compound eyes and evolutionary trade-offs between visual acuity and body size constraints.

## 2. Materials and Methods

### 2.1. Wasp Community in Ficus Semicordata and Sample Preparations

*Ficus semicordata*, a functionally dioecious species, maintains an obligatory mutualism with its exclusive pollinator *Ceratosolen gravelyi*, which serves as a host of four non-pollinating wasps from Sycophaginae (*Sycophaga cunia*) and Pteromalidae (*Philotrypesis dunia*, *Apocrypta* sp., and *Sycoscapter trifennensis*). The reproductive cycle is initiated when *C. gravelyi* locates receptive trees and enters male figs for offspring development and female figs for pollination. Following this pioneer species, *P. dunia*, *S. cunia*, *Apocrypta* sp., and *S. trifennensis*, which are usually sympatric, sequentially oviposit into figs at progressive developmental stages over ca. 30 days (Figure 1). All offspring reach sexual maturity simultaneously by developmental stage D, and their oviposition schedules are staggered. Adult females of all five species exhibit fully developed wings, eyes, and coordinated emergence, but males have reduced eyes, atrophied antennae, and an apterous morphology and usually perish within the fig cavity.

We collected newly matured stage D figs of *F*. *semicordata* (male wasps emerge from galls while females remain within the galls) from Xishuangbanna Tropical Botanical Garden, Menglun, China (21°41′ N, 101°25′ E). Figs were enclosed in nylon mesh bags to permit natural wasp emergence. Newly emerged individuals were anesthetized at −20 °C for 20 min. For ultrastructural observations, samples were divided into two groups: (1) intact wasps fixed in 2% glutaraldehyde for SEM, and (2) decapitated specimens (including both natural light- and dark-adapted (red light) treatments) were immersed in a mixed fixative solution (2.5% glutaraldehyde, 4% paraformaldehyde, 0.1 mol/L phosphate-buffered saline) for TEM. All samples were maintained at 4 °C for 3 days.

### 2.2. Scanning Electron Microscopy (SEM)

Following ultrasonic cleaning, 20 selected female specimens underwent sequential dehydration using a graded ethanol series (50%, 75%, 80%, and 95% for 10 min each and absolute alcohol for 20 min twice). The specimens were then subjected to critical point drying using a Quorum K850 dryer (Quorum, Ashford, Kent, UK). The dried samples were mounted by eye orientation (left/right) and sputter-coated with a gold film for 30 s using a Cressington 108 coater (Cressington, Chalk Hill, Watford, UK) to enhance electrical conductivity. One specimen was bisected along its head midline with a fine blade to enable a detailed mapping of facet distribution and surface area. Standard imaging was performed using a Zeiss Sigma300 field emission scanning electron microscope (Carl Zeiss, Jena, Germany) operating at an acceleration voltage of 20 kV. However, for the representative replica selected for detailed spatial mapping of facet size and distribution, the sample was deliberately tilted to optimize facet diameter measurement and counting. Finally, facet size and distribution patterns were analyzed using a custom R script (Version 4.4.1; https://cran.r-project.org/, accessed on 5 April 2025).

### 2.3. Transmission Electron Microscopy (TEM)

Following fixation, the heads were rinsed three times in phosphate-buffered saline (PBS; 0.1 M, pH 7.2) for 7 min per wash and postfixed with 1% osmium tetroxide (OsO_4_) (Ted Pella, Altadena, CA, USA) in PBS at 4 °C for 2 h. Subsequently, the samples were washed three times with double-distilled water (ddH_2_O) (7 min for each wash), followed by dehydration with a graded ethanol series (as previously described) and 5 min in acetone. The specimens were then embedded in Epon 812 resin (SPI-Chem, West Chester, PA, USA) and polymerized at 60 °C for 48 h. After post-polymerization, uniform serial sections were prepared using a Leica EM UC7 ultramicrotome (Leica, Wetzlar, Germany), including semithin sections (800 nm thickness) and ultrathin sections (60 nm thickness). Ultrathin sections were mounted on copper grids, dual-stained with 2% uranyl acetate and lead citrate, and examined using a JEM-1400 Plus transmission electron microscope (JEOL Ltd., Tokyo, Japan) operating at 80 kV.

### 2.4. Optometric Measurements and Calculations

All morphological analyses were performed using Image J software Version 1.53 (National Institutes of Health, Bethesda, MD, USA). SEM micrographs were utilized to quantify dorsal-ventral (height) and antero-posterior (width) ocular dimensions as well as the diameters of facets and ocelli. Total ommatidial counts per eye (right vs. left) were determined from corneal replicas. For TEM analyses, longitudinal sections provided measurements of ommatidial length, corneal thickness, cone length, rhabdom length, and corneal facet dimensions; transverse sections enabled the quantification of rhabdom diameter (distal and proximal regions), microvilli diameter, and morphological characteristics (shape and size) of pigment granules in primary pigment cells (PPCs), secondary pigment cells (SPCs), and retinular cells [13,15]. The radius of eye curvature was calculated at the maximum eye width using a circular measurement tool over the eye surface in the image analysis software. Rhabdom diameter measurements were conducted at both distal and proximal rhabdom extremities. Retinula cell counts followed standardized *Drosophila* numbering, while morphology (R8 positioning, basal cells, axonal projections) was evaluated according to Fischer et al. (2011) [13]. Analyses included at least three female *C. gravelyi* eyes per treatment group (SEM/TEM; details in Table 1). All TEM images represent natural light treatment unless stated otherwise.

To evaluate the optical characteristics of female *C. gravelyi* eyes, various key parameters were calculated from anatomical measurements (See Table 2 for details). These calculations followed Makarova et al. (2015) [15], except for the F-number, which used a revised formulation (See Appendix A for details).

## 3. Results

### 3.1. General Morphology of the Eye

In female *C. gravelyi*, a pair of convex reddish compound eyes is symmetrically positioned on the side of the head (Figure 2A,B). These oval-shaped eyes measure 137.8–164.7 µm in dorsoventral length (eye height) and 188.0–210.5 µm in anteroposterior length (eye width), and they contain 228–263 ommatidia with facet diameters (corner-to-corner distance) ranging from 9.3 to 13.5 µm (Figure 2C; Table 1). Central facets display a typical hexagonal geometry and are regularly arranged, while peripheral facets progressively transition from pentagonal lattices to strongly curved cuticular structures, which are particularly pronounced at the eye margins (Figure 2D). The largest facets were observed in the anteroventral region, and the smallest facets were observed in the posteroventral zone (Figure 3). The corneal surface features smooth interommatidial spaces averaging 0.66 μm in width, with 13–19 interommatidial hairs (6.5–21.6 µm long, 1.0–1.7 µm thick) that are predominantly localized in the mid-inferior ocular region (Figure 2C,D). Notably, ommatidial counts exhibit bilateral asymmetry within individuals (e.g., left eyes: 248, 247, 228; right eyes: 262, 263, 256, respectively), and SEM analysis indicated that this stems from variation in spatial constraints during ommatidial differentiation (Figure 3). Additionally, three ocelli are dorsoposteriorly positioned; a circular median ocellus (35.0 µm diameter) is flanked by two elongate lateral ocelli (33.2 µm diameter) (Figure 2B; Table 1).

### 3.2. Fine Structural Organization of the Eye

Each ommatidium comprises two elements: the distal dioptric apparatus and the proximal photoreceptive retinular. The dioptric apparatus includes a biconvex, multilayered cornea and a tetrapartite crystalline cone, both of which are surrounded by primary and secondary pigment cells. The photoreceptive retina contains light-sensitive retinular cells and screening pigment cells. All retinular cells converge to form a fused, cone-shaped rhabdom, which is distally connected to the crystalline cone. Figure 4 represents semi-schematic illustrations of the ommatidium structure in female *C. gravelyi*. The average length of an ommatidium is 54.3 ± 4.3 µm (Figure 5A; Table 1), and the interommatidial angle between adjacent ommatidia measures 9.3°. In general, the rhabdoms appear uniformly arranged across the eye (Figure 5B); however, occasional irregularly shaped rhabdoms are observed in peripheral regions.

#### 3.2.1. Dioptric Apparatus

The transparent, biconvex corneal lens features a smooth external surface with outer and inner radii of curvature measuring 7.1 µm and 8.7 µm (Figure 5A and Figure 6A; Table 1), respectively. Each corneal facet has a central thickness of approximately 5.7 µm and comprises approximately nine horizontally stacked microfibrils that progressively thicken from the distal to proximal regions (Figure 6A). Corneal thickness is minimal in the interommatidial areas. The subcorneal layer lies immediately beneath the cornea, appearing light-colored, loosely arranged, and randomly oriented without discernible pattern, with a maximum thickness of 0.38 µm (Figure 6A,B).

The eucone-type crystalline cone measures 12.7 µm in average length and 8.9 µm in maximum width and comprises four cone cells (also termed Semper cells) (Figure 6C–E; Table 1). Each cone cell contains a large nucleus positioned in its distal region, although no other organelles were observed (Figure 6C). The cone matrix is densely packed with ribosomes and particles less than 2 nm in diameter, likely glycogens (Figure 6E,G). In the longitudinal section, the cone exhibits a conical shape that gradually narrows proximally before transitioning into a slender crystalline tract (Figure 6F). Within this tract, the cone cell processes interweave with each other and form an interlaced labyrinth (Figure 6G). The crystalline tract, which is approximately 5.20 µm in width and 2.51 µm in length, directly connects to the rhabdom’s distal end (Figure 6F,G). Cone cell projections extend to the rhabdom and terminate at the basal matrix (Figure 4 and Figure 6G).

#### 3.2.2. Pigment Cells

Primary pigment cells (PPC), also known as pigmented corneagenous cells, are consistently present in pairs in each ommatidium (Figure 4 and Figure 6A,D–F). They symmetrically envelop both the crystalline cone and the crystalline tract, and their proximal ends contact the distal tips of the retinular cells (Figure 6F,G). The nucleus of each PPC is a large, ellipsoidal structure positioned between the cornea and the proximal end of the crystalline cone (Figure 6A). It connects to the innermost subcorneal layer via hemidesmosomes. The cytoplasm of PPC contains numerous mitochondria and spherical pigment granules; the latter are highly electron-dense and measure approximately 0.62 µm in maximum diameter (Figure 6D,E; Table 1). Proximally, the PPC forms a sheath around the crystalline cone and extends narrow, pigment-free projections. These projections, only a few micrometers long, extend into the spaces between the retinular cells near the rhabdom (Figure 6G). The PPCs of adjacent ommatidia are closely opposed to each other, but they leave gaps elsewhere to accommodate secondary pigment cells (SPC) (Figure 6A,D,F).

SPCs comprise six cells positioned in the spaces between the PPC of adjacent ommatidia and the cornea (Figure 6A,D,F). These cells are restricted to this region and do not extend to the basal matrix; thus, they cannot be classified as interommatidial pigment cells. In the cone region, SPCs are relatively narrow but broaden between the crystalline tract and the distal tip of the rhabdom, tapering again as they extend proximally. Their nuclei exhibit highly condensed chromatin and measure 3.4 μm in transverse section, and they are located at the base of the crystalline cone. The cytoplasm of the SPC also contains abundant mitochondria and spherical, electron-dense pigment granules, the largest reaching ca. 0.57 μm in diameter (Figure 6D; Table 1).

#### 3.2.3. Retinula Cells and Rhabdom

The rhabdom, measuring approximately 35.7 µm in length, spans from the proximal end of the crystalline cone to just above the basal membrane (Figure 4; Table 1). In longitudinal sections of *C. gravelyi* eyes, the nuclei of retinular cells are distributed along nearly the entire length of the retinular cells, from the distal rhabdom region to the lower third, rather than being confined to one plane (Figure 7A,E–I). Notably, no retinular cell nuclei were observed in the proximal third adjacent to the basal membrane (Figure 7J). Furthermore, in any given transverse plane, the nuclei of two adjacent retinular cells from neighboring ommatidia are rarely positioned at the same level (Figure 7E–I). This alternating nuclear arrangement within a single ommatidium and between neighboring ommatidia allows for more efficient use of the limited space.

In the distalmost region of the *C. gravelyi* ommatidium, the rhabdom attains its maximum diameter (averaging 2.4 µm; Table 1). At this level, up to three retinular cell nuclei can be observed within a single horizontal plane (Figure 5B, Figure 7E and Figure 8A,B). Progressing proximally, as the ommatidial radius decreases, the number of nuclei per transverse plane and horizontal plane correspondingly diminishes. The centrally fused rhabdom is formed along its entire length by eight regular retinular cells. However, within the distal two-thirds of the rhabdom, a ninth retinular cell (R9, classified as irregular) emerges (Figure 7B). This R9 displaces R8 and contributes a small rhabdomere to the rhabdom structure. This positional shift causes three adjacent rhabdomeres to have an identical microvillar alignment, thus forming a rhabdomeric triplet (Figure 7C and Figure 8A). Adjacent retinular cells are interconnected by adhesive junctions featuring desmosomes. Mitochondria are frequently observed clustered near the cell membrane at these junction sites (Figure 5B, Figure 7D and Figure 8A,B). The microvilli constituting the rhabdomeres originate from two pairs of opposing retinular cells (totaling four cells). The microvilli from one pair are uniformly aligned in a single direction. The microvilli from the other pair are parallel to each other but oriented.

Orthogonal to those of the first pair (Appendix A). Consequently, transverse sections through the rhabdoms at this level reveal microvilli oriented in just two orthogonal directions (Figure 7C and Figure 8A). Individual microvilli measure approximately 57.2 nm in diameter (Table 1). Extending proximally alongside the retinular cells, four cone cell projections reach down to the basal membrane (Figure 6G, Figure 7B,E–J, Figure 9A and Figure 10A,B). These projections serve to separate the pairs of retinular cells.

The rhabdom is encircled by cisternae of smooth endoplasmic reticulum (ER), which is collectively termed the perirhabdomeric (PER, Figure 4, Figure 6G, Figure 7C and Figure 8A,B). The structure of this specialized PER system varies significantly with the state of light adaptation. During light adaptation, the palisade arrangement of the PER forms a reduced, pigment-free annular zone around the rhabdom, with a maximum width measuring only 0.21–0.84 µm (*N* = 10; Figure 7C and Figure 8A). Conversely, under dark-adapted conditions, PER expands considerably, reaching a maximum width of 1.24–2.71 µm (*N* = 10; Figure 8B). In addition to the PER, the retinular cells contain ovoid pigment granules. These granules have an average longitudinal diameter of 0.47 µm (Table 1). Aside from their location, the granules in retinular cells show strong similarities to pigment granules PGP and PGS in both external morphology and electron density (Figure 6A, Figure 7A,B,E and Figure 8A). Analysis of longitudinal sections further revealed that these pigment granules show no preferential alignment along the ommatidial longitudinal axis. This random orientation suggests no correlation between granule position and available space within the ommatidium of female *C. gravelyi*.

Further proximally, all retinular cells progressively taper along the rhabdom and transition into axons approximately 1 µm above the basal matrix (Figure 7B, Figure 9A and Figure 10A,B). Therein, retinular cells R3, R4, and R7 undergo early volumetric reduction, whereas the opposing cells R2 and R6 maintain their full length, extending from the distal to proximal extremities of the rhabdom (Figure 7B, Figure 9B and Figure 10B). These latter cells (R2, R6) are consequently the longest retinular cells in the ommatidium (Figure 9C–E). Additionally, cells R1 and R5 are positioned diametrically opposite from each other within the ommatidium, yet share identical microvillar orientations. Cell R9 follows this sequence and has a similar axonal transformation. Near the proximal terminus of the rhabdom, R2 and R6 enlarge roughly and contain a few pigment granules (Figure 9D). Ultimately, the axons of all retinular cells, including those of the atypical R9, converge into discrete bundles of nine fibers that traverse the basal matrix (Figure 9D and Figure 10B).

#### 3.2.4. Basal Matrix

The basal matrix, measuring approximately 198.9 nm in thickness, forms a critical boundary layer between the retina and lamina, demarcating the optical compartment from the neuropil (Figure 5A, Figure 7B and Figure 10A; Table 1). This matrix is perforated by regularly spaced circular apertures that permit the passage of ommatidial axon bundles, which are each ensheathed by glial cells. Each bundle contains nine fibers: two centrally positioned large-diameter fibers flanked by three smaller fibers on each lateral side, with an additional small fiber occupying the central axis of the bundle (Figure 10B,C). Abundant mitochondria are present in each axon (Figure 10A,C). Additionally, the interommatidial space in the basal matrix allows tracheoles to project distally into the optical region (Figure 10A,B). The region directly beneath the basal matrix houses large nuclei and electron-dense pigment granules, which potentially have screening functions.

### 3.3. Optical Properties

Following the methodology described in Makarova et al. (2015) [15], eleven key optical parameters of female *C. gravelyi* eyes are summarized in Table 2. Calculations using lens curvatures (P_1_, P_2_, P_3_) and a total lens power (P_l_ = P_1_ + P_2_ + P_3_) of 0.073 yielded a focal length of 13.7 µm and an image focal length of 18.5 µm (See Appendix A for details). These results indicate that lens power is predominantly governed by the outer surface curvature (Table 2), with the calculated focal plane positioned near the distal tip of the rhabdom. The F-number, derived from the focal length and facet diameter measurements, was determined to be 1.1 for *C. gravelyi*, which is lower than values reported for *T. evanescens* and *A. flavipes*, but marginally higher than that of *M. mymaripenne* (Table 2). Further analysis revealed an acceptance angle Δρ_rh_ of 10.0° and a half-width blur circle (Δρ_1_) of 2.4°. Notably, the optical sensitivity (S_w_) reached 0.26 µm^2^/sr, substantially exceeding values observed in the three comparative parasitoid wasp species.

## 4. Discussion

Comprehensive morphological analyses of hymenopteran compound eyes have been performed for taxa in numerous common families such as Apoidea, Vespidae, and Halictidae [2,5,14,43,44]. However, ultrastructural research on parasitoid wasps is scarce [2]; the lack of studies on parasitoid wasps is particularly significant given their estimated global diversity of approximately 500,000 to 1,000,000 species [25]. The compound eyes of only three species, *M. mymaripenne* [15], *T. evanescens* [13], and *A. flavipes* [15], have been analyzed to date. Our findings provide novel insights into the ultrastructure of the compound eyes of an additional parasitoid species, which enhances our comparative understanding of vision within this ecologically significant group.

### 4.1. Structural Features of the Eyes of C. gravelyi

#### 4.1.1. External Morphology

The external morphology of compound eyes, particularly overall shape, facet number, and facet size, may be essential for effective function in variable light environments. The eye of *C. gravelyi* has an oblong and narrow shape, and the anteroposterior length (eye width: 201.3 µm) exceeds the dorsoventral length (eye height: 147.6 µm) (Figure 2C; Table 1). This morphological pattern is consistent with observations reported for daytime active parasitoid wasps exceeding 0.3 mm in total body size [13,15], but is in contrast to the ocular dimensions of the smallest known thrips egg parasitoid, *M. viggianii*, which displays a short and wide eye [9]. These features are influenced by environmental factors, biological traits, diurnal activity rhythms, and sex [13,15,29,45]. A recent large-scale analysis of insect compound eyes by Makarova et al. (2022) [2] revealed allometric relationships between ommatidial traits and body size: Ommatidia number y = 56.89x^1.46^ and Ommatidia diameter y = 9.84x^0.36^. Notably, *C. gravelyi* (body size > 1.5 mm), which is morphometrically similar to small hymenopterans in Makarova’s dataset, has eye dimensions and facet diameters comparable to other fig wasps, which corroborates these scaling patterns. However, critical gaps remain in parasitoid wasp research. Makarova’s segmented regression analysis highlights this discontinuity and reveals divergent scaling slopes between wasps at over 1.5 mm and those below 1.5 mm. Additionally, insects within the 2–5 mm body size range exhibit specialized adaptations [1,23], which further emphasizes the need for systematic morphological investigations of parasitoid wasps varying in size.

Furthermore, both facet diameter and lens curvature influence sensitivity and spatial resolution [16], yet facet diameter cannot be reduced indefinitely. Makarova et al. (2022) [2] demonstrated 4- to 30-fold variation in facet diameters across insects of varying sizes, ranging from large to the smallest known species. However, parasitoid wasps exhibit minimal interspecific variation in this trait. For instance, *C. gravelyi* (body length 2.74 mm) in this study has a facet diameter of 11.8 µm, whereas miniaturized parasitoid species (body length < 0.4 mm) such as *T. evanescens* [13], *M. viggianii* [9], *A. flavipes* [15], and *Kikiki huna* [46] show facet diameters of 6.39 µm (female), 8.1 µm, 8.1 µm, and 7.65 µm (SEM-measured), respectively. These measurements reveal facet diameters far below theoretical limits, even in extreme cases of miniaturization such as *M. viggianii* [9]. In general, the eyes of parasitoid wasps are often notably convex, which is a structural constraint resulting from their reduced radius of curvature. This morphology, exemplified by the curved facets in the dorsal rim area of *C. gravelyi* (Figure 2D), likely enhances the visual field by expanding angular sensitivity, even under severe miniaturization [2,23].

#### 4.1.2. Dioptric Apparatus

In nearly all parasitoid wasps [8,13,15,46], the corneal lens exhibits a smooth external surface characterized by distinct outer and inner radii of curvature. A reduction in the outer corneal radius of curvature is correlated with an increased radius of curvature across the ommatidial surface, producing more convex facets. Here, we demonstrate that lens power is primarily determined by the anterior lens surface (Table 2), and the outer facet radius is the critical factor for achieving short focal lengths (f). This optical requirement is physiologically essential for miniature species, where optimization of the focal length is evolutionarily constrained [47]. To further shorten the focal lengths, additional reductions in the curvature radius would necessitate a proportional decrease in facet diameter. However, this imposes functional trade-offs given that smaller facets exacerbate light diffraction and reduce interommatidial angles, thereby diminishing ommatidial sensitivity. The examined eyes circumvent diffraction limitations, as the blur circle width remains smaller than the rhabdom’s acceptance angle. Furthermore, gradual variation in facet size from the eye’s center to periphery (Figure 3) may reflect adaptive modifications to enhance light-capture efficiency during navigation [17,45]. Surprisingly, a well-developed subcorneal layer always appears beneath the cornea in *C. gravelyi* females (Figure 6A,B). However, this structure is absent in previously studied parasitoid wasps [13,15] but is ubiquitously present in nonparasitic hymenopterans [20,45]. Phylogenetic comparisons further reveal its convergent evolution in the nocturnal moth *Macrosoma* sp. [48] and *Grapholita molesta* [24]. The maximum thickness of this layer within individual ommatidia is inversely related to the ambient light intensity [24]. This supports the hypothesis that the subcorneal layer functions as a photomechanical adaptation for low-light vision, particularly critical for crepuscular behaviors such as dawn dispersal in fig wasps emerging from natal figs [49]. The eucone crystalline cone of *C. gravelyi* measures approximately 12.7 µm in length, which is significantly longer than those of smaller parasitoid wasps such as *A. flavipes* (10.0 µm) [15], *T. evanescens* (7.88 µm) [13], and *M. mymaripenne* (5.3 µm) [15]. However, these dimensions are not allometrically correlated with body size compared with crystalline cones in other hymenopterans [2,5,45]. More significantly, we identified for the first time an interlaced labyrinth at the distal region of the crystalline cone cell, which connects directly to the distal end of the rhabdom (Figure 6G). This ubiquitous morphology may potentially enhance light capture efficiency due to its large contact surface area. Additionally, it may facilitate rapid host-finding by pollinators like *C. gravelyai*, enabling them to locate available figs efficiently [49].

#### 4.1.3. Pigment Cells and Pigment Granules

Two types of pigment cells, PPC and SPC, are commonly present across Hymenoptera. However, the position and number of SPCs encircling the PPC show considerable variation within this order. For example, in miniaturized parasitoid wasps, SPCs terminate proximally at the crystalline cone level and lack distal extensions, a morphology completely distinct from that observed in large hymenopterans like vespid species [2,43]. Reported SPC counts also vary: *T. evanescens* ommatidia contain either 5 [26] or 6 SPCs [13], while *M. viggianii* exhibits up to 24 SPCs distributed across 29 ommatidia [9]. Makarova et al. (2025) [9] reveal regional specialization: central ommatidial rows are encircled by 4 SPCs, while peripheral rows contact only 2 or 3. We confirm *C. gravelyi* possesses 6 SPCs per ommatidium (Figure 6C), consistent with *T. evanescens* [13]. However, comprehensive 3D reconstruction of *C. gravelyi* is warranted due to reported discrepancies between 2D and 3D ultrastructural data in *T. evanescens*. Pigment granule dimensions (e.g., size, shape, orientation) vary between parasitoids and other hymenopterans. In this study, granule diameter in *C. gravelyi* shows no significant divergence from larger species, supported by 3D reconstructions in *T. evanescens* [26] and *M. viggianii* [9]. In contrast, miniature parasitoids exhibit adaptive modifications in spatial organization: their granules are ellipsoidal rather than spherical as in larger hymenopterans [2,17,22,43]. Unlike reports for miniature parasitoids [13,15], Coleoptera [7], and Lepidoptera [23], *C. gravelyi* granules in this study lack parallel alignment to the ommatidial axis, suggesting insufficient constraints. Finally, retinular cells in miniature parasitoids (*M. mymaripenne*, *A. flavipes*) have significantly higher pigment granule packing density (13 and 5 per cell) compared to *C. gravelyi* (4–5), likely an evolutionary adaptation to spatial limitations [15].

#### 4.1.4. Retinula Cells and Rhabdom

Like other miniature parasitoids [13,15], the apposition compound eyes of *C. gravelyi* feature a distal rhabdom extension directly connected to the crystalline cone. Its distal rhabdom diameter measures 2.4 µm, exceeding the 2 µm threshold and thus functioning as a light guide [16]. Regardless of whether their rhabdoms act as waveguides or lightguides, the perirhabdomeric structures observed in these parasitoid species mediate critical transitions between dark and light adaptation. This mechanism likely enhances photon capture efficiency in apposition eyes, optimizing visual performance across varying light intensities [50]. While nine retinular cells are typical in hymenopterans, the ninth cell initiates more proximally in *C. gravelyi* and miniature parasitoids (*M. mymaripenne*, *T. evanescens*, *A. flavipes*), likely due to their shortened rhabdoms compared to larger relatives [9,13,15]. These spatial modifications, including retinular cell contributions, are conserved across hymenopterans, indicating evolutionary stability. However, miniaturization drives structural adaptations, causing divergence from ancestral states [2,13,15,18,43]. A key adaptation in all studied parasitoids is the interlocking arrangement of retinular cells between neighboring ommatidia, characterized by staggered nuclear positioning. This optimizes tissue packing efficiency within and between ommatidia. Volumetric constraints also drive size reduction or loss of non-critical organelles (e.g., nuclei, mitochondria, ER), enhancing spatial economy in extreme miniaturization [2]. In the smallest species, *M. mymaripenne* (<0.3 mm), spatial constraints from shortened ommatidia and increased rhabdom diameters drive partial nuclear displacement into inter-ommatidial spaces between primary pigment cells. This redistribution, unlike the conserved ommatidial proportions in larger parasitoids (>0.3 mm) [7,9,13,15], is a compensatory adaptation to maintain functional demands like orientation, navigation, and host recognition.

### 4.2. Optical Features and Ecological Demand of the Eyes of C. gravelyi

Theoretical analyses demonstrate that optimizing compound eye functionality in a given species requires coordinated adjustments among three core components: the dioptric apparatus, photoreceptive rhabdom, and essential organelles, which are shaped by ecological demands [16,27,47]. Increase in rhabdom diameters to improve light capture in dim environments, as observed in species such as *C. gravelyi* (10° acceptance angle; body > 2 mm) and *M. mymaripenne* (22.2°; body size < 2 mm) (Table 2), reduce organelle space and necessitate shorter lengths of the dioptric apparatus. While shortened focal lengths coupled with broad rhabdoms increase acceptance angles, there are trade-offs in these adaptations. *C. gravelyi* achieves 2.6 times higher light sensitivity than *A. mellifera* (0.1 µm^2^/sr) due to rhabdom enlargement [22], whereas smaller parasitoids (*A. flavipes*, 0.04 µm^2^/sr; *T. evanescens*, 0.06 µm^2^/sr) show only half *A. mellifera*’s sensitivity [13,15]. Comparative analyses of acceptance angles across hymenopterans (e.g., intertidal ants, honeybees, parasitoids) reveal inherent constraints in widening acceptance angles [16,22,45]. However, since adequate photon capture is fundamental for detectable signals [2,19], sensitivity outweighs other parameters in miniature eyes, exemplified by photon-prioritizing eyeless cave spiders [51]. Consequently, miniaturization drives non-allometric reductions in ommatidial count, similar to larger species where facet number scales linearly with body size [2,3,5,23,27]. Reduced body size primarily yields fewer facets per eye. A key strategy involves maintaining optimized numbers of smaller facets to preserve resolution: *C. gravelyi* possesses twice the ommatidia of *T. evanescens* [13] and 8.5 times more than *M. mymaripenne* [15], reflecting size-dependent scaling of ommatidial density under physical constraints.

V-parameter analysis indicates female *C. gravelyi* exhibits trichromatic light sensitivity, overlapping with *T. evanescens* in second and third order modes [13]. This suggests enhanced sensitivity in blue, violet, and ultraviolet wavelengths (<500 nm, <400 nm, and far UV), dominating rhabdom light absorption [16,50,52]. Consistent with phototactic behavior and genomic studies of two *Ceratosolen* species [37,38], *C. gravelyi* shows higher light sensitivity than *M. mymaripenne*, *T. evanescens*, and *A. flavipes* (Table 2), likely stemming from V-parameter optimization for trichromacy. Adapted for precise host fig detection, *C. gravelyi* females are diurnally active, relying on high light sensitivity [18,49,53]. Their perirhabdomeric adaptations to light/dark treatments likely facilitate navigating the dramatic contrast between the bright fig exterior and dim cavity interior. Conversely, species with small eyes, low ommatidial counts, and reduced optical parameters are not inherently visually limited. The ant *Leptothorax albipennis* navigates effectively with only 60 ommatidia using visual landmarks [54]. Similarly, fig wasps like *C. gravelyi* rely heavily on olfactory and chemical signals for host location. Neural adaptations, including multimodal sensory integration [40], can offset restricted visual input, enabling complex behaviors even with sparse ommatidia [9,40,54,55]. Future research combining RNA-seq and CRISPR/Cas9 to analyze gene expression and function will advance the understanding of visual cue detection, processing, and their multimodal behavioral integration.

## 5. Conclusions

We investigated the fine structural and optical characteristics of the compound eyes in adult female *C. gravelyi*, the exclusive pollinator of *F. semicordata*, using scanning and transmission electron microscopy. The apposition compound eyes contain 228–263 ommatidia per eye, exhibiting asymmetric distribution between left and right eyes. Each ommatidium comprises a biconvex corneal lens covering a tetra-partite eucone crystalline cone; proximal cone cells reveal an interlaced labyrinth. Pigment cells surround each ommatidium, with numerous pigment granules and mitochondria localized in both pigment cells and retinular cells. Nine retinular cells constitute each unit: eight photoreceptors (R1–R8) form the rhabdom from the cone base to the basal matrix, while a ninth cell replaces R8 in the apical third. Optical calculations indicate the eyes exhibit predominantly trichromatic light sensitivity and are optimized for diurnal activity in bright environments. These findings, combined with future research on gene expression and function, will advance the understanding of multimodal sensory integration (e.g., visual, olfactory, and chemical signals) in the fig–wasp mutualism and elucidate how specialized eye morphology supports ecological niche specialization.

## Figures and Tables

**Figure 1 insects-16-00682-f001:**
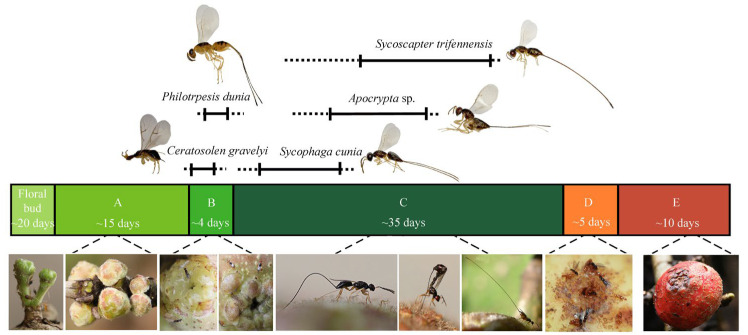
Oviposition patterns of five parasitoid wasp species across five developmental stages (A–E) of *Ficus semicordata* figs. Solid lines represent oviposition timing for the top 50% of individuals in each species, while dotted lines indicate the oviposition activity of the remaining individuals. Top inset panels display morphological differences among the five wasp species, and bottom inset panels illustrate both fig surfaces across stages and species-specific oviposition behaviors corresponding to different fig stages.

**Figure 2 insects-16-00682-f002:**
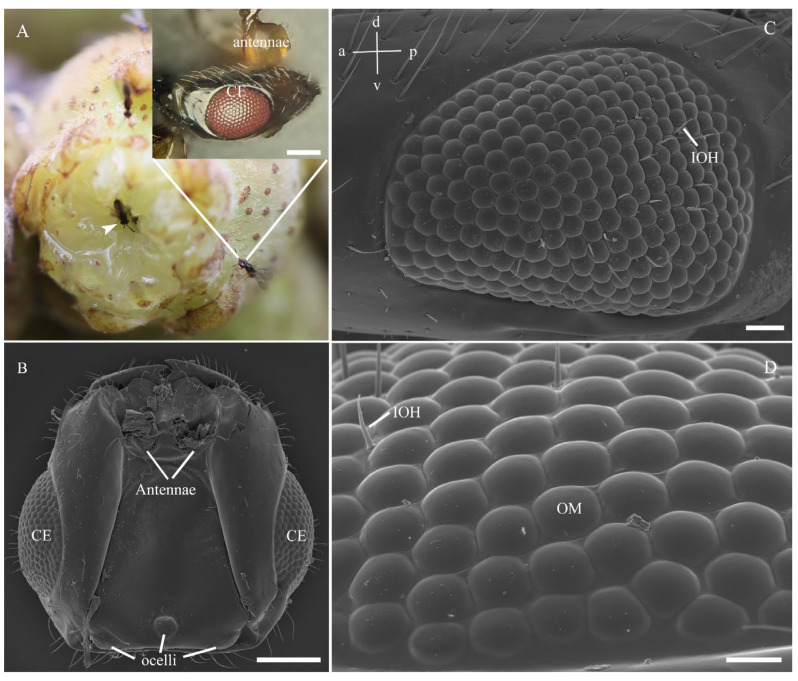
External morphological features of the compound eyes in female *C. gravelyi*. (**A**) Females exhibit a distinct and reddish-colored compound eye (CE, top right); the entry point of female wasps into the fig through the apical ostiole (arrowhead). (**B**) A pair of convex eyes is positioned symmetrically on the sides of the head. (**C**) The eyes display an overall oval shape, with sparse interommatidial hairs (IOH) predominantly localized in the mid-inferior region (a, anterior; p, posterior; v, ventral; d, dorsal). (**D**) External details of the ommatidia (OM) show the smooth, curved facets characteristic of the dorsal rim area. Scale bar: (**A**,**B**) and inset = 100 µm; (**C**) = 20 µm; (**D**) = 10 µm.

**Figure 3 insects-16-00682-f003:**
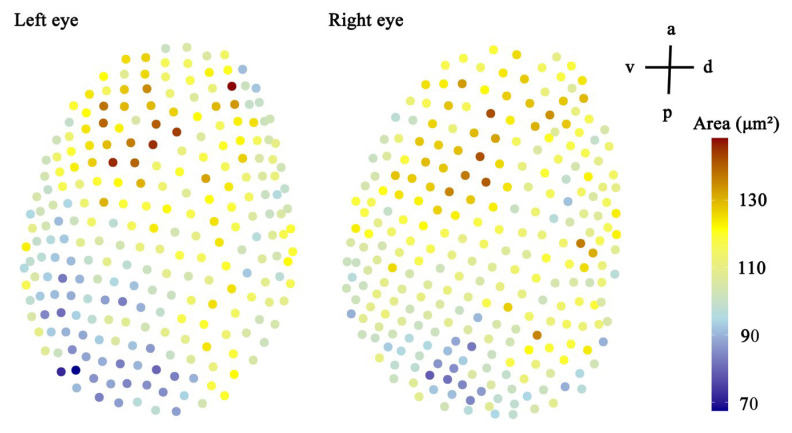
With anatomical orientation indicated in the top-right corner (a, anterior; p, posterior; v, ventral; d, dorsal).

**Figure 4 insects-16-00682-f004:**
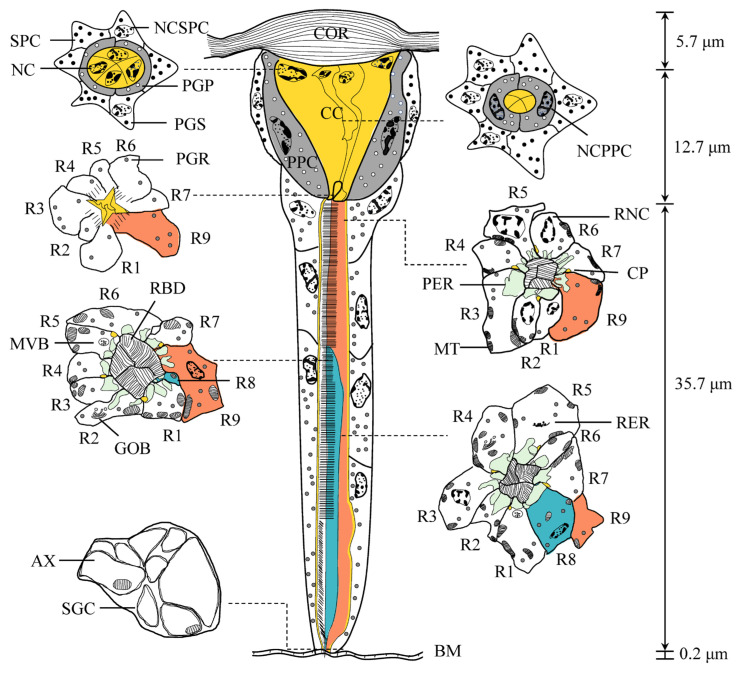
Semi-schematic illustrations of a single ommatidium from female *C. gravelyi*, showing longitudinal (middle) and corresponding transverse sections (dashed lines). Key structures and abbreviations: AX, axon; BM, basal matrix; CC, cone cell (yellow); COR, cornea; GOB, Golgi body; PGP, PGS and PGR, pigment granule present in primary/secondary pigment cell and retinular cell, respectively; MT, mitochondria; MVB, multivesicular bodies; NC, nucleus; NCPPC, nucleus of primary pigment cell; NCSPC, nucleus of secondary pigment cell; PER, perirhabdomeric (green); PPC, primary pigment cell (gray); R1–9, retinular cell numbers 1–9, the labeled cyan and orange regions represent the position of retinular cell 8 and 9, respectively; RBD, rhabdom; RER, rough endoplasmic reticulum; RNC, retinular cell nucleus; SGC, sheal of glial cell; SPC, secondary pigment cell.

**Figure 5 insects-16-00682-f005:**
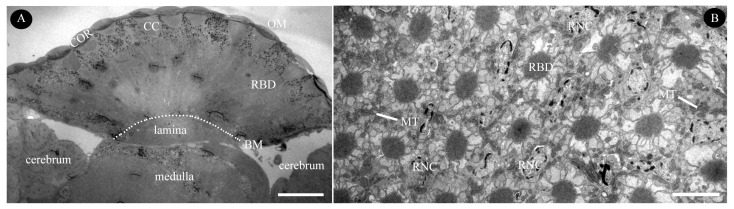
An overview of the compound eyes in female *C. gravelyi*. (**A**) Longitudinal section of the eye in the anterior-right plane, with the cerebrum positioned adjacent to the eye. The dotted line denotes the boundary of the basal matrix. (**B**) Transverse section of the eye in the distal region, displaying rhabdom and associated cellular components. BM, basal matrix; CC, cone cell; COR, cornea; MT, mitochondria; OM, ommatidia; RBD, rhabdom; RNC, retinular cell nucleus. Scale bar: (**A**) = 20 µm; (**B**) = 5 µm.

**Figure 6 insects-16-00682-f006:**
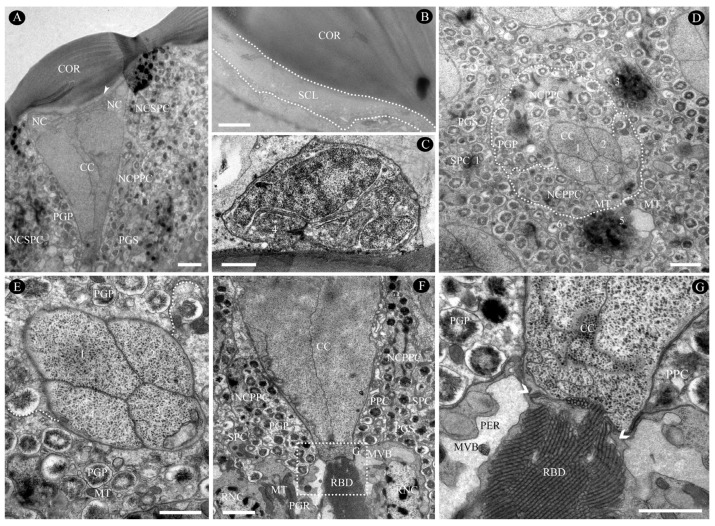
Ultrastructure of the dioptric apparatus in the compound eye of female *C. gravelyi*. (**A**) Longitudinal section through the complete dioptric apparatus. The arrowhead indicates a subcorneal layer positioned between the corneal lens and crystalline cone. (**B**) Magnification reveals the subcorneal layer (dotted line) as light-colored, loosely arranged, and unoriented. (**C**) A transverse section through the tip of a cone cell reveals four nuclei. (**D**) Transverse section through the distal third of the cone cell, showing four representative cone cells with numerical labels. The dotted lines demarcate the boundary between primary pigment cells. Cone cells and associated pigment cells are numbered clockwise for orientation. (**E**) Magnification reveals four cone cells and one primary pigment cell, the latter containing numerous pigment granules and several mitochondria. (**F**) Longitudinal section of the cone cell’s proximal region, showing structural details. (**G**) High-magnification image of the cone-rhabdom conjunction (white dotted box in (**F**)), revealing an interlaced labyrinth. The arrowhead indicates a cone cell process extending to the rhabdom. CC, cone cell; COR, cornea; PGP, PGS, and PGR, pigment granule present in primary/secondary pigment cell and retinular cell, respectively; MT, mitochondria; MVB, multivesicular bodies; NC, nucleus; NCPPC, nucleus of primary pigment cell; NCSPC, nucleus of secondary pigment cell; PER, perirhabdomeric; PPC, primary pigment cell; RBD, rhabdom; RNC, retinular cell nucleus; SCL, subcorneal layer; SPC, secondary pigment cell. Scale bar: (**A**,**D**,**F**) = 2 µm; (**B**,**C**,**E**,**G**) = 1 µm.

**Figure 7 insects-16-00682-f007:**
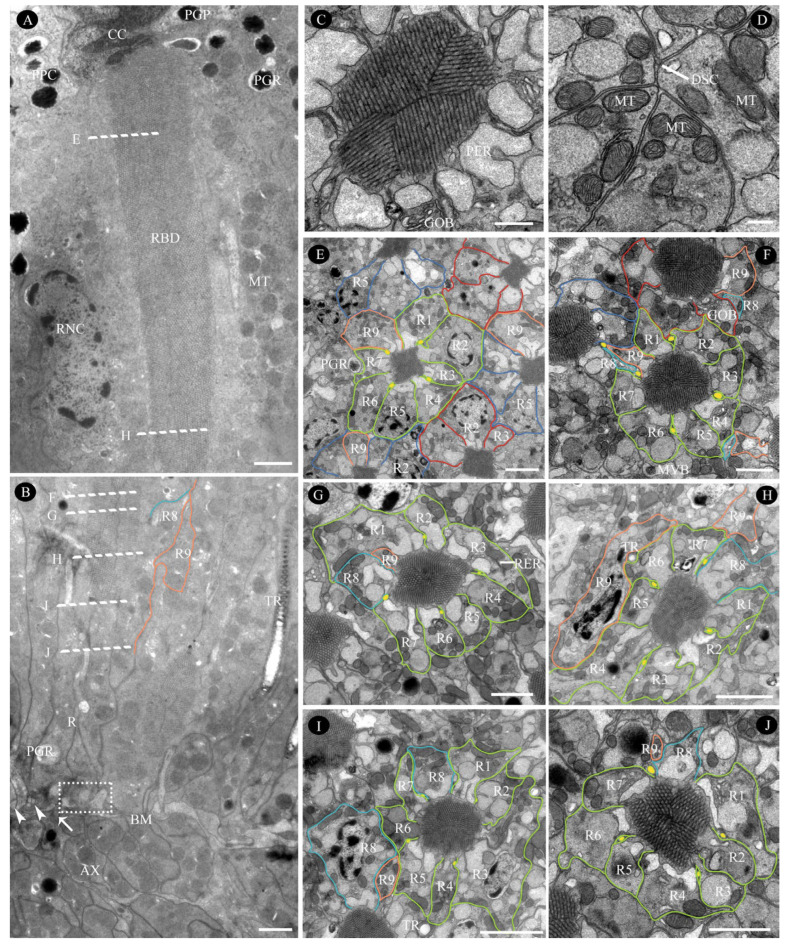
Ultrastructure of retinular cells and rhabdom in the compound eye of female *C. gravelyi*. (**A**) Longitudinal section through the distal third of the rhabdom. (**B**) Longitudinal section of the proximal two-thirds of the rhabdom, demonstrating the overlapping region between retinular cells 8 and 9. The cone cell projection (arrowhead) and axon (arrow) penetrating the basal matrix are indicated. The dotted box highlights four cone cell projections and two retinular cell processes extending to the basal matrix. (**C**) Transverse section of the rhabdom showing vertically oriented microvilli. (**D**) Retinula cell adhesive junctions, featuring desmosomes and mitochondria clustered near the cell membrane. (**E**–**J**) Serial transverse sections of the rhabdom at levels corresponding to the dashed lines in (**A**,**B**). Retinula cells located in the center of the photo are labeled lime-green; other retinular cells were assigned red and steel-blue. AX, axon; BM, basal matrix; CC, cone cell; DSC, desmosomes; GOB, Golgi body; PGP, PGR, pigment granule present in primary pigment cell and retinular cell, respectively; MT, mitochondria; MVB, multivesicular bodies; PER, perirhabdomeric; PPC, primary pigment cell; R1–9, retinular cell numbers 1–9, the labeled cyan and orange regions represent the position of retinular cell 8 and 9, respectively; RBD, rhabdom; RCN, retinular cell nucleus; RER, rough endo-plasmic reticulum; RNC, retinular cell nucleus; TR, tracheole. Scale bar: (**A**,**F**) and (**J**) = 1 µm; (**B**,**E**,**G**–**I**) = 2 µm; (**C**) = 500 nm; (**D**) = 200 nm.

**Figure 8 insects-16-00682-f008:**
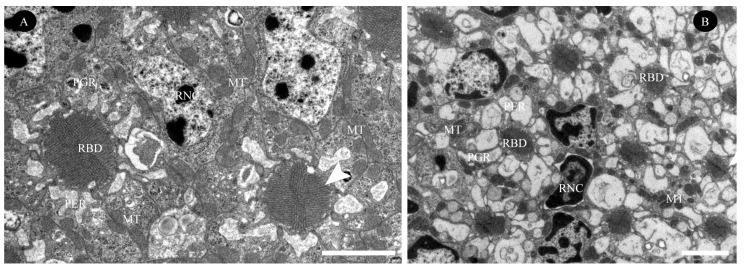
Transverse section through the distal third of the rhabdom showing the perirhabdomeric changes in natural light—(**A**) and dark-adapted (**B**) treatment. PGR, pigment granule present in retinular cell; MT, mitochondria; PER, perirhabdomeric; RBD, rhabdom; RNC, retinular cell nucleus. Scale bar: (**A**,**B**) = 2 µm.

**Figure 9 insects-16-00682-f009:**
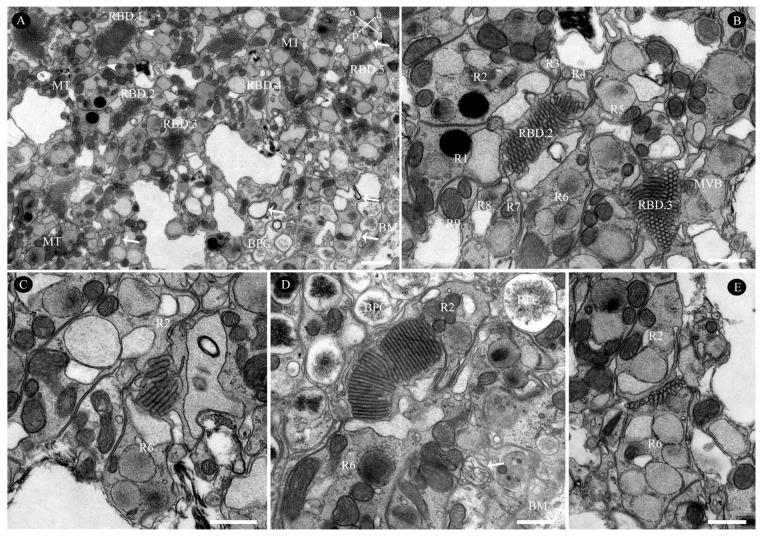
Transverse sections through the basal region of the rhabdom in the compound eye of female *C. gravelyi*. Arrowheads indicate cone cell projections adjacent to rhabdomeres, while arrows show axons penetrating retinular cells. (**A**) Transverse section through the basal region of the rhabdom reveals differentially oriented microvilli and sequential reduction among rhabdomeres (labeled RBD.1–5; o: outer, i: inner, a: anterior, p: posterior). (**B**) Magnification of rhabdomeres 2 and 3 shows early volume reduction in retinular cells R3, R4, and R7, contrasting with the relative stability of opposing cells R2 and R6. (**C**) Magnification of rhabdomere 4 reveals only retinular cells R2 and R6 extending to the proximal extremities of the rhabdom. (**D**) Near the proximal terminus of the rhabdom, cells R2 and R6 show significant enlargement. (**E**) At the distal tip of the rhabdom, retinular cells R2 and R6 are nearly absent. BM, basal matrix; BPC, basal pigment cell; MT, mitochondria; MVB, multivesicular bodies; R1–9, retinular cell numbers 1–9; RBD.1–5, rhabdom 1–5. Scale bar: (**A**) = 1 µm and (**B**–**E**) = 500 nm.

**Figure 10 insects-16-00682-f010:**
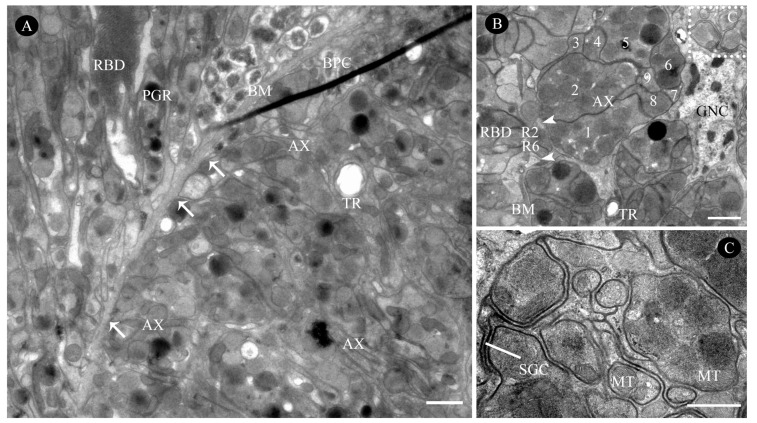
Ultrastructure of the basal matrix in the compound eye of female *C. gravelyi*. (**A**) Oblique section through the proximal-most retinular cell terminals, demonstrating axonal penetration of the basal matrix via ommatidial cartridges (arrow). (**B**) Transverse sections of axon bundles are surrounded by glial cell sheaths, each containing a prominent glial nucleus. The rhabdom lies opposite the axons, showing retinular cells 2 and 6. A cone cell projection (arrowhead) terminates at the distal end of the basal matrix. Numbers indicate each bundle contains nine axons, but this ordering does not match the retinular cell sequence. (**C**) High magnification image of a single axon bundle from panel B (the dotted box), showing ultrastructural details. AX, axon; BM, basal matrix; BPC, basal pigment cell; GNC, glial nucleus; SGC, sheal of glial cel; MT, mitochondria; PGR, pigment granule present in retinular cell; RBD, rhabdom; TR, tracheole; Scale bars: (**A**,**B**) = 1 µm; (**C**) = 500 nm.

**Table 1 insects-16-00682-t001:** Histological and optical parameters of the compound eyes in female *C. gravelyi*. Data are shown as mean ± standard deviation.

Structural Elements	Parameters	Unit	*N*	Average	Range (Min–Max)
Compound eyes	eye height	µm	20	147.6 ± 5.9	137.8–164.7
	eye width	µm	20	201.3 ± 6.0	188.0–210.5
	facet number	-	11	247 ± 14	228–263
	eye radius	µm	3	90.4 ± 5.9	86.4–97.4
Ommatidia	facet diameter	µm	150	11.8 ± 0.7	9.3–13.7
	length	µm	30	54.3 ± 6.8	46.8–59.9
	interommatidial angle	deg	30	9.3 ± 0.9	7.7–12.5
Cornea	outer lens radius	µm	20	7.1 ± 0.6	5.6–8.3
	inner lens radius	µm	20	8.7 ± 0.9	7.7–10.8
	maximum thickness	µm	50	5.7 ± 0.8	4.1–7.9
	number of chitin layers	-	20	9.0 ± 1.0	8–10
Crystalline cone	length	µm	50	12.7 ± 2.3	9.3–18.4
	distal diameter	µm	50	8.9 ± 1.2	6.4–11.3
Pigment granule	PPC diameter *	µm	150	0.62 ± 0.11	0.4–1.0
	SPC diameter *	µm	150	0.57 ± 0.12	0.3–0.8
Rhabdom	length	µm	20	35.7 ± 4.3	29.1–43.9
	distal diameter	µm	20	2.4 ± 0.2	2.0–2.8
	proximal diameter	µm	12	0.69 ± 0.1	0.5–0.9
	microvillus diameter	nm	150	57.2 ± 7.5	41.1–81.6
	PG diameter *	µm	150	0.47 ± 0.10	0.2–0.7
Interommatidial hairs	number	-	20	15 ± 2	13–19
	length	µm	30	13.0 ± 3.9	6.5–21.6
	basal diameter	µm	30	1.4 ± 0.2	1.0–1.7
Basal matrix	thickness	nm	20	198.9 ± 63.3	128–354
Ocellus	diameter	µm	15	35.0 and 33.2	25.5–42.8

*, marks the largest diameter, as pigment granules are oval in shape.

**Table 2 insects-16-00682-t002:** Optical parameters of female *C. gravelyi* were calculated from direct eye measurements, and comparative data for three parasitoid wasp species were obtained from previously published studies by Makarova et al. (2015) [15] and Fischer et al. (2011) [13].

Parameters *	*C. gravelyi*	*M. mymaripenne*	*T. evanescens*	*A. flavipes*
Body size (mm)	2.74	0.2	0.3–0.4	0.45
P_1_	0.064	0.133	0.098	0.076
P_2_	0.012	0.037	0.016	0.021
P_3_	−0.003	−0.009	−0.002	−0.004
P_I_	0.073	0.161	0.112	0.094
f (µm)	13.7	6.2	8.9	10.7
f′ (µm)	18.5	8.4	12.0	14.4
F-number	1.1	0.8	1.4	1.3
Δρ_rh_ (°)	10.0	22.2	8.1	4.8
Δρ_1_ (°)	2.4	3.5	4.5	3.5
S_w_ (μm^2^/sr)	0.26	0.23	0.06	0.04

* All parameters are defined below. P_1_, outer surface of lens; P_2_, inner surface of lens; P_3_, neutralizing power; P_I_, lens power; f, focal length; f′, image focal length; Δρ_1_, Airy disk half-width; Δρ_rh_, acceptance angle; S_w_, optical sensitivity.

## Data Availability

The original contributions presented in this study are included in the article/Appendix A. Further inquiries can be directed to the corresponding authors.

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
