# Peer review of "Fine Structure and Optical Features of the Compound Eyes of Adult Female *Ceratosolen gravelyi* (Hymenoptera: Agaonidae)"

_insects, 2025, doi:10.3390/insects16070682_

Round 1
Reviewer 1 Report
Comments and Suggestions for Authors
This is a traditional study of compound eye morphology of small of fig wasp Ceratosolen gravelyi (Hymenoptera: Agaonidae). The manuscript contains a description of eye morphology illustrated by TEM (unfortunately weak quality). The discussion builds on the comparison the results with the data on miniature hymenopterans in light of how miniaturization affects the compound eye structure. This study complements the available data on scaling of eyes in insects, confirms the trends but not give any new results.
General remarks:
In general, most of the TEM figures are bad illustrate the descriptions, probably due not successful sections. The using of SEM for internal morphology of eyes is questionably. The illustrations cause more questions than applied to solve. Directly most of them do not illustrate the described results.
It is a bit fancily that authors use the outdated methodology of replicas with nail polish (Ribi et al, 1989) to count and measure the facet number, considering that they have the opportunity to make the SEM of wasp heads. The SEM data give more precisely results for these measurements, without the deformations.
The most of figure captions (probably except of figure 2 and 4) lack the abbreviation description. And most of the abbreviations directly on figures are not visible.
As stated above, the discussion largely consists of repetition of previously published data, and the results of this work add little new to the discussion. The discussion should be rewritten around new results, rather than as a retelling of previous work by other authors.
Minor remarks along the text:
Line 19: “neurons (retinula cells)” no need to simplify, just retinula cells
Line 21: ommatidial core – what is it? Maybe authors mean the basal matrix
Line 38: the same “ommatidail core”
Materials and methods: Need a section on the principle of retinal cell numbering.
The scan of the internal structure of the ommatidia seems pointless. All discussion and interpretation of the images is questionable.
Line 222: on lateral (side? ) of the head
Line 223: It seems that there is a mismatch in the drawing. At fig 2C dorsal/ventral is mixed up with anterior/posterior. If so, some sections should be rewrited based on the correct orientation. For example, in discussion 458-459.
The caption to Fig.2 C.gravelyi should be highlighted in italics -> C. gravelyi
Fig.4 SPC boundaries are absent. Do they have projections to the BM or not? What are the dots in the picture along the entire ommatidium?
Fig. 5 lacks explanation of abbreviations for the figure
All sections need to be redone taking into account the exclusion of scan illustrations, on which nothing is visible, and the replacement of some illustrations fig 9, on which what is described is also not visible)
Line 286-288: pores are not visible in the picture
Fig.6. A - the boundaries between the cells are subjective.
Fig.7 there are no legends in the figure captions. The abbreviations on the figures are not visible (D- PAL) and what is pal? The shape of the fragment D is not the same as lined in B.
Line 317: Dotted lines delimit cells, not organelles.
Line 327-329: there should be any symbols or arrows on the picture to clarify which of these are projections of PPC
Line 340: there should be mitochondria in SPC!
Line 363: banded pattern should be illustrated on longitudinal sections. It is not a common organization. A banded rhabdom organization, in which layers of microvilli are arranged perpendicular to each other, is known in decapod crustaceans and in few insect species (Meyer-Rochow, 1972; Makarova, Polilov, 2018; Yang et al., 2024). The occurrence of a banded rhabdom in decapods and in some hexapod species may result from a convergence in relation to some special function in light reception.
Line 375: figures 6,7 do not illustrates the high density near the ER
Line 378: the alignment of pigment granules do not visible at figure 9A.
Fig. 8 there is no explanation of the symbols in the captions
Line 389: what is “central rhabdom”?
Line 396: does not show the rhabdomeric triplet
Line 409: change “grandules” to “granules”
Fig 9 There is no explanation of the Abbr in the figure captions. Nothing is clearly visible on the images. The bundle of axons needs to be marked so that the boundaries are clear.
In the text, either M. mymaripenne or M. viggianii is linked to the same references. For example, line 453 M. mymaripenne (8,9,15) and line 484 M. viggianii (9,15). M.viggianii should be indicated everywhere based on the revision Fusu, Polaszek & Polilov 2022
Line 497: change “cpmstraints” to “constraints”
Line 544-545: and how it consistent with findings on C. gravelyi ?
Line 695: “few pigment granules” what is it mean? There are not a few…
Author Response
|
Response to Reviewer 1 Comments
|
||
|
1. Summary |
|
|
|
Thank you for your valuable feedback and suggestions regarding our manuscript (insects-3665047). We have fully incorporated all of your recommendations. All modifications made in response to your comments are clearly highlighted in red within the resubmitted manuscript files. We have also specifically addressed the points raised by the reviewers as follows:
(1) TEM Image Descriptions: We have carefully re-examined all TEM images and have revised their descriptions based on the typical characteristics observed in each image. We believe these updated descriptions resolve the previous points of confusion.
(2) SEM Images of Internal Compound Eye Morphology: We acknowledge that the quality of these specific SEM images was insufficient and raised questions. Consequently, we have removed these images. We maintain that their removal does not affect the overall results presented in the text, as the key findings are clearly substantiated by the TEM data. Further, we have also added new TEM images in the resubmitted files.
(3) Updated Methodology for Facet Analysis: We have discontinued the outdated replica methodology using nail polish (referencing Ribi et al., 1989) for facet counting and distribution pattern analysis. Instead, we have reverted to using SEM to observe the compound eye surface. All relevant images and associated measurement data are now provided in the Supplementary Materials (see files: Facet number and diameter.zip and measurement data.xlsx).
(4) Image Abbreviations: We have meticulously reviewed all figure abbreviations throughout the manuscript to ensure accuracy and clarity. Detailed definitions for all abbreviations are now provided, and we confirm their correctness in the resubmitted files.
(5) Discussion Section: We have thoroughly revised the Discussion section. The revised text provides a clearer summary and focuses the discussion more concisely on the interpretation of our specific results, reducing reliance on comparisons with previous work by other authors where appropriate. We believe this results in a more accurate and focused discussion.
Finally, we appreciate the opportunity to improve our manuscript and hope these revisions fully address the reviewers' concerns. Please find our detailed revisions in the resubmitted files.
|
||
|
2. Questions for General Evaluation |
Reviewer’s Evaluation |
Response and Revisions |
|
Does the introduction provide sufficient background and include all relevant references? |
Can be improved |
We have incorporated essential research background detailing the unique role of the fig wasp visual system. These modifications are highlighted in red within the resubmitted manuscript files. |
|
Are all the cited references relevant to the research? |
Yes |
While reviewers did not request changes to references, we conducted a thorough review of all cited sources central to our research. |
|
Is the research design appropriate? |
Must be improved |
We have replaced the outdated replica methodology using nail polish (Ribi et al., 1989) with SEM observations, which provide more precise results for facet counting and distribution pattern analysis. |
|
Are the methods adequately described? |
Yes |
We optimized the experimental methods within the Materials and Methods section to better reflect the study's integrated design. |
|
Are the results clearly presented? |
Must be improved |
We recognize that previous confusion may have stemmed from either the presentation of images/result descriptions or the inadvertent omission of a key figure in the initial submission. All such issues have now been resolved: errors were corrected, missing image restored (Figure 9), and result descriptions comprehensively rewritten. These revisions are labelled red in the resubmitted files. |
|
Are the conclusions supported by the results? |
Must be improved |
In accordance with the new description, we have corrected all of the conclusions in the resubmitted files. |
|
3. Point-by-point response to Comments and Suggestions for Authors |
||
|
Comments 1: Line 19: “neurons (retinula cells)” no need to simplify, just retinula cells
|
||
|
Response 1: Thank you for your feedback. In the resubmitted files, we have replaced "neurons" with retinula cells at line 19. Additionally, we have corrected some specific errors in the abstract section on lines 16-17. Please review these changes.
|
||
|
Comments 2: Line 21: ommatidial core – what is it? Maybe authors mean the basal matrix. |
||
|
Response 2: We apologize for the spelling error; as correctly noted by the reviewer, this should be basal matrix at line 20.
|
||
|
Comments 3: Line 38: the same “ommatidail core”. |
||
|
Response 3: Thank you for your feedback. We have made the necessary revisions accordingly at line 38.
Comments 4: Materials and methods: Need a section on the principle of retinal cell numbering. Response 4: Agree, we have added the following description at lines 198-200: "Retinula cell counts followed standardized Drosophila numbering, while morphology (R8 positioning, basal cells, axonal projections) was evaluated according to Friedrich et al. (2011) [13] ". Please review it in the resubmitted files. Thanks.
Comments 5: The scan of the internal structure of the ommatidia seems pointless. All discussion and interpretation of the images is questionable. Response 5: We fully agree and have removed these images. Their exclusion does not affect the core conclusions, as key findings are robustly supported by the TEM data. We have additionally included new TEM images in the resubmitted manuscript. Comments 6: on lateral (side? ) of the head. Response 6: We agree that a descriptive error was present and have revised line 210 in accordance with the reviewer’s recommendation.
Comments 7: It seems that there is a mismatch in the drawing. At fig 2C dorsal/ventral is mixed up with anterior/posterior. If so, some sections should be rewrited based on the correct orientation. For example, in discussion 458-459. Response 7: To more accurately represent head orientation, we have annotated Figure 2C. We attribute previous misinterpretations to the unique morphological adaptations of pollinating fig wasps. These insects exhibit an elongated head morphology (see new Figure 2C in resubmitted files), an adaptation for traversing the fig ostiole, with their dorsal surface bearing two rows of sclerites. This provides a reliable diagnostic feature (see Figure 1 below). Additionally, facet size and distribution patterns in female C. gravelyi (Figure 3 in resubmitted files) offer further identification cues. Collectively, this evidence confirms the accuracy of Figure 2C. |
||
|
Figure 1 Head morphology and orientation in a female pollinating fig wasp. Note the subtle anterior extension of the compound eye compared to posterior regions (arrowhead).
Comments 8: The caption to Fig.2 C.gravelyi should be highlighted in italics -> C. gravelyi Response 8: We agree and have implemented this correction at line 227, with the relevant text now clearly labeled. Thank you for your feedback.
Comments 9: Fig.4 SPC boundaries are absent. Do they have projections to the BM or not? What are the dots in the picture along the entire ommatidium? Response 9: Completely agree. We have redrawn Figure 4 in new files and corrected all errors, including the cone cell projection and black dotted circles. The latter were previously labeled as errors in the previous figure. We sincerely apologize for my omission.
Comments 10: Fig. 5 lacks explanation of abbreviations for the figure Response 10: Yes, we have added the abbreviations relevant to Figure 5. Furthermore, we have included all abbreviations involved in all figures within the new text.
Comments 11: All sections need to be redone taking into account the exclusion of scan illustrations, on which nothing is visible, and the replacement of some illustrations fig 9, on which what is described is also not visible) Response 11: Yes. The original Figure 9 illustrated changes in retinula cells and rhabdoms within the compound eye of female C. gravelyi. Regrettably, during manuscript revision, the corresponding author inadvertently deleted a key image set. This omission exacerbated confusion in our initial submission. We have now included a new Figure 9 in the resubmitted files that accurately presents these morphological changes. We sincerely apologize for this oversight and respectfully request the reviewer's examination of the revised descriptions in the updated text. Thank you for your attention to this matter.
Comments 12: Line 286-288: pores are not visible in the picture Response 12: Agree. We believe this issue was caused by the SEM images of internal compound eye morphology. We have removed this image set from the new files. Thank you.
Comments 13: Fig.6. A - the boundaries between the cells are subjective. Response 13: Yes, we have removed this image set from the new files. Thank you.
Comments 14: Fig.7 there are no legends in the figure captions. The abbreviations on the figures are not visible (D- PAL) and what is pal? The shape of the fragment D is not the same as lined in B. Response 14: Yes, we have now regrouped these images in the updated text (see new Figure 6F-G). We believe that this new image set can better demonstrate the accuracy and performance of the dioptric apparatus in C. gravelyi. Additionally, we have explained why Figure 7D was chosen in the previous text, as the longitudinal section provides a clearer view compared to the magnified version of the same image. This new image demonstrates the same result as the previous Figure 6D, as shown below: it displays the cone cell's proximal region (left) and its structural details (right). Thank you.
Figure 2 Longitudinal section of the cone cell's proximal region and its basal structural details.
Comments 15: Line 317: Dotted lines delimit cells, not organelles. Response 15: Please see the above reply. This revised image set provides an accurate representation of the dioptric apparatus in C. gravelyi, enhancing both clarity and analytical value. Thank you. Comments 16: Line 327-329: there should be any symbols or arrows on the picture to clarify which of these are projections of PPC Response 16: Yes, the previous description raised some questions. Based on the new image set, we have redescribed the results in the updated manuscript. Please review it at lines 312-316.
Comments 17: Line 340: there should be mitochondria in SPC! Response 17: We fully agree and have implemented this correction in the revised manuscript, supplemented with new supporting images. Thank you.
Comments 18: Line 327-329: there should be any symbols or arrows on the picture to clarify which of these are projections of PPC Response 18: We have incorporated these symbols into the updated manuscript's images. Thanks.
Comments 19: Line 340: there should be mitochondria in SPC! Response 19: Yes, the SPC have typical mitochondria, which are labeled in Figure 6D of the updated file. Please review it. Thanks.
Comments 20: Line 363: banded pattern should be illustrated on longitudinal sections. It is not a common organization. A banded rhabdom organization, in which layers of microvilli are arranged perpendicular to each other, is known in decapod crustaceans and in few insect species (Meyer-Rochow, 1972; Makarova, Polilov, 2018; Yang et al., 2024). The occurrence of a banded rhabdom in decapods and in some hexapod species may result from a convergence in relation to some special function in light reception. Response 20: Regarding the fused rhabdom of female C. gravelyi, there was a significant error in our previous description. The key cause was misclassifying the longitudinal direction as transverse, which led to an incorrect characterization of the fused rhabdom as a band rhabdom. We have identified this artificial mistake and corrected these descriptions in the updated manuscript. Furthermore, we have revisited and discussed this correction in the discussion section. Please review these changes. Thank you.
Comments 21: Line 375: figures 6,7 do not illustrates the high density near the ER Response 21: We identified some errors in the previous manuscript and have added a new image set that more clearly illustrates the changes between the perirhabdomeric and pigment granules under both light and dark adaptation. Please review these updates in the revised manuscript. Thank you.
Comments 22: Line 378: the alignment of pigment granules do not visible at figure 9A. Response 22: Please refer to our response to comment 21. Thank you.
Comments 23: Fig. 8 there is no explanation of the symbols in the captions Response 23: Yes, we have corrected these errors in the revised manuscript. Please review the updated Figure 7. Thank you.
Comments 24: Line 389: what is “central rhabdom”? Response 24: Yes, we recognized that the original description could be misinterpreted. This has been corrected to “Retinula cells in the central region of the micrograph are labeled lime-green; adjacent retinula cells are assigned red and steel-blue”. Please review this revision on lines 363-364 of the revised manuscript.
Comments 25: Line 396: does not show the rhabdomeric triplet Response 25: We have added a new image (Figure 7C in the revised manuscript) to support this description. Please review this update in the revised manuscript. Thank you.
Comments 26: Line 409: change “grandules” to “granules” Response 26: “Grandules" is now corrected to "granules" in the text. Thank you.
Comments 27: Fig 9 There is no explanation of the Abbr in the figure captions. Nothing is clearly visible on the images. The bundle of axons needs to be marked so that the boundaries are clear. Response 27: We have corrected the abbreviation error throughout and marked the axon bundle (dotted box in Figure 7B of the revised manuscript). The updated figure better illustrates our findings. Additionally, we have included a new Figure 9 and 10 providing further evidence supporting these results. We believe these revisions address all reviewer concerns. Please evaluate the updated manuscript. Thank you.
Comments 27: In the text, either M. mymaripenne or M. viggianii is linked to the same references. For example, line 453 M. mymaripenne (8,9,15) and line 484 M. viggianii (9,15). M.viggianii should be indicated everywhere based on the revision Fusu, Polaszek & Polilov 2022 Response 27: We acknowledge this error, which arose because studies on this parasitoid wasp were conducted by the same author at different times. To resolve this conflict, we have cited the most directly relevant reference in the revised manuscript. Please review this correction in the updated document. Thank you.
Comments 28: Line 497: change “cpmstraints” to “constraints” Response 28: “cpmstraints " is now corrected to " constraints " in the text. Thank you.
Comments 29: Line 544-545: and how it consistent with findings on C. gravelyi ? Response 29: The original manuscript may contain an inaccurate description. We alway observe the subcorneal layer of C. gravelyi females lying immediately beneath the cornea. This has been clarified in the revised manuscript (lines 523-524): The subcorneal layer consistently appears immediately beneath the cornea in C. gravelyi females. Please verify this correction. Thank you.
Comments 30: Line 695: “few pigment granules” what is it mean? There are not a few… Response 30: This inaccurate description has been corrected to “Pigment cells surround each ommatidium, with pigment granules and mitochondria localized in both pigment cells and retinula cells“ (revised manuscript, lines 636-637). Please verify this revision. Thank you. |
||
|
4. Response to Comments on the Quality of English Language |
||
|
Point 1: The English could be improved to more clearly express the research |
||
|
Response 1: We understand that the English quality in the original manuscript could be enhanced. We have addressed and improved all language-related issues through a professional Language Service Company in China (https://www.liwenbianji.cn/). If you require certification for the language service, we will provide it upon request. Thank you. |
||
|
|
||
|
|
||

Reviewer 2 Report
Comments and Suggestions for Authors
This study uses light/scanning/transmission electron microscopy to provide a detailed description of the compound eye morphology and structure of pollinating fig wasps. Indeed, the importance of vision in pollinating wasps' search for specialized hosts and other behavioral activities has been overlooked. This study provides a preliminary basis for further exploration of the importance of vision in pollinating wasps' behavior and life activities. The specific evaluation opinions are as follows:
The introduction provides a detailed explanation of the diversity and structure of insect compound eyes, but some research background on the specific role of the visual system of fig wasps in symbiotic relationships can be added to better introduce the purpose and significance of this study. For example, how pollinating wasps visually evaluate the availability of host syconia at receptive phase. In addition, I think the author may have missed a more important role of vision, which is that mature and mated female wasps in ripe syconia need to use compound eyes to feel the light emitted by the fly holes dug by male fig wasps.
The Materials and Methods section is described in detail, but specific information about sample size and statistical analysis can be added, especially in the tables of results (Tables 1 and 2) where the statistical sample size of each characteristic parameter should be listed.
The discussion provides a detailed analysis of C. gravelly's compound eye characteristics, but the content is relatively difficult to understand. It is suggested to link these analyses with the behavior and life characteristics of pollinating wasps, such as “Regardless of their rhabdomeric configuration as wave guides or light guides, the palisade ER structures (Figures 4, 8E–J) observed across these parasitoid species mediate critical transitions between dark and light adaptation states. This mechanism likely enhances the photon capture efficiency of apposition eyes, which optimizes visual performance across varying light intensities.” This may related to that the pollinating wasps can fly out of the mature syconia and then enter the receptive syconia, and requiring adaptation to different light intensities. In addition, why do pollinating wasps have strong sensitivity to tricolor light? What specific behavioral characteristics can blue, purple, and ultraviolet (<500 nm,<400 nm, and far ultraviolet) be associated with? The author added comparisons with other species, but the overall feeling is not very clear. It should follow the principle of first discussing similarities (what common behaviors are reflected), and then discussing specificity (why is there specificity?).
The conclusion section can propose future research directions.
Author Response
|
Response to Reviewer 2 Comments
|
||
|
1. Summary |
|
|
|
Thank you for your valuable feedback and suggestions regarding our manuscript (insects-3665047). We have fully incorporated all of your recommendations. All modifications made in response to your comments are clearly highlighted in red within the resubmitted manuscript files. We have also specifically addressed the points raised by the reviewers as follows:
(1) Background description: We acknowledge a gap in existing research regarding pollinating fig wasps, particularly concerning the behavioral ecology where mated females exhibit contrasting states when exiting fig cavities (dark to light) versus re-entering figs (light to dark). This suggests females may use compound eyes to detect light transmitted through the fig's apical opening. Supporting this hypothesis, our new Figure 8 demonstrates significant structural changes in perirhabdomeric regions between light-adapted and dark-adapted conditions. We have incorporated this background in the revised manuscript (lines 101-107).
(2) Materials and Methods section: We have addressed the lack of sample size, even though some values were already listed in Table 1. However, due to sexual dimorphism and the fact that only females disperse and search for available hosts, we did not analyze or compare these results from the importance of ecological functions. While we could also analyze the changes under light versus dark conditions, there are no significant differences except for the perirhabdomeric volumes. Obviously, this problem comes from our description in material and method section in original manuscript. We have corrected it in Section 2.4 and redescribed it on lines 198-202.
(3) The discussion section: We have completely rewritten this section following the reviewer's guidance, with particular emphasis on the visual ecology and pollination functions of fig wasps in the revised manuscript. Additionally, we have focused on our specific findings by removing some unrelated results from the new files. Please review these changes in the update manuscript.
(4) Further research: Based on this advice, we have added future research directions. Specifically, combining RNA-seq and CRISPR/Cas9 techniques will allow us to analyze gene expression and function, thereby advancing our understanding of visual cue detection, processing, and their multimodal behavioral integration.
|
||
|
2. Questions for General Evaluation |
Reviewer’s Evaluation |
Response and Revisions |
|
Does the introduction provide sufficient background and include all relevant references? |
Can be improved |
We have incorporated essential research background detailing the unique role of the fig wasp visual system. These modifications are highlighted in red within the resubmitted manuscript files. |
|
Are all the cited references relevant to the research? |
Yes |
While reviewers did not request changes to references, we conducted a thorough review of all cited sources central to our research. |
|
Is the research design appropriate? |
Yes |
While no experimental design changes were requested by reviewers, we also optimized the methodology description to strengthen reproducibility and alignment with results. |
|
Are the methods adequately described? |
Yes |
We have also updated our methods in the revised manuscript based on feedback from other reviewers. |
|
Are the results clearly presented? |
Yes |
We have also revised some unclear results in the original manuscript, particularly regarding the omission of a key figure (Figure 9 in the revised manuscript). These changes are labeled in red within the resubmitted files. |
|
Are the conclusions supported by the results? |
Can be improved |
In accordance with the new description, we have corrected all of the conclusions in the update manuscript. |

Reviewer 3 Report
Comments and Suggestions for Authors
Dear authors,
The article provides basic information on the morphology, ultrastructure and optical properties of the compound eyes of a species of pollinating fig wasp (Agaonidae). The data presented are interesting, and the methodology was planned correctly.
General remarks:
However, a more detailed introduction to the differences between the eye constructions of the three species in the same family (Agaonidae) is entirely missing. It would be clearer and more sensible to compare the ultrastructural characters in a new table (expand Table 2 and connect with the suplmentary tables ) for these species. This would improve the understanding of most of the discussion.
Figures 7 and 8 need to be more closely linked to the text, and more characters need to be indicated in the figures according to the description of the same characters in the text.
Some of the results are difficult to understand due to weak image documentation support.
Additional comments are included in the PDF file.

Author Response
|
Response to Reviewer 3 Comments
|
||
|
1. Summary |
|
|
|
Thank you for your valuable feedback and suggestions regarding our manuscript (insects-3665047). We have fully incorporated all of your recommendations. All modifications made in response to your comments are clearly highlighted in red within the resubmitted manuscript files. We have also specifically addressed the points raised by the reviewers as follows:
(1) Detailed description in Introduction: We have incorporated research background detailing the unique role of the fig wasp visual system. These detailed modifications are highlighted in red within the resubmitted manuscript files. Please refer to lines 101–107 in the updated manuscript for these changes. Thank you for your feedback and consideration.
(2) Expand Table 2: According to your feedback, we have expanded the results presented in Table 2. However, due to this expansion, there is now a potential conflict between the material and methods section and the revised Table 2 regarding some descriptions. To resolve this, we have simplified the calculations in line with the methodology described by Makarova et al. (2015). For further details, please refer to lines 203–206 and the annotations in Table 2 within the updated manuscript.
(3) Figures 7 and 8 need to be more closely linked to the text: We have regrouped these TEM images to more accurately describe their performance. Please see the updated Figures 6, 7, and 9 in the revised manuscript.
|
||
|
2. Questions for General Evaluation |
Reviewer’s Evaluation |
Response and Revisions |
|
Does the introduction provide sufficient background and include all relevant references? |
Yes |
We have incorporated essential research background detailing the unique role of the fig wasp visual system. These modifications are highlighted in red within the resubmitted manuscript files. |
|
Are all the cited references relevant to the research? |
Yes |
While reviewers did not request changes to references, we conducted a thorough review of all cited sources central to our research. |
|
Is the research design appropriate? |
Yes |
While no experimental design changes were requested by reviewers, we also optimized the methodology description to strengthen reproducibility and alignment with results. |
|
Are the methods adequately described? |
Yes |
We have also updated our methods in the revised manuscript based on feedback from other reviewers. |
|
Are the results clearly presented? |
|
We have also revised some unclear results in the original manuscript, particularly regarding the omission of a key figure (Figure 9 in the revised manuscript). These changes are labeled in red within the resubmitted files. |
|
Are the conclusions supported by the results? |
Can be improved |
In accordance with the new description, we have corrected all of the conclusions in the update manuscript. |
|
3. Point-by-point response to Comments and Suggestions for Authors |
||
|
Comments 1: Line 4: and * |
||
|
Response 1: Yes, there were errors in the author sections of the original manuscript, which we have corrected in this revised version. Thank you.
Comments 2: Line 54: among strains? Response 2: Yes, we acknowledge that the previous phrasing was unclear. To enhance clarity, we have replaced "among strains" with "closely related species". We believe this revised description more accurately conveys the intended information, based on our cited reference.
Comments 3: Line267, 272:What is the difference between PGP and SPC below? Response 3: We now distinguish PGP/PGS granules solely by their location in primary (PPC) vs. secondary pigment cells (SPC), as they are otherwise identical. PPC and SPC are differentiated by their positional relationship to cone cells: PPC envelops the cone cell body, while SPC borders the PPC. Please refer to Figure 6 in the revised manuscript for the illustration of cell boundaries. Thank you.
|
||
|
Comments 3: Line 295: it is unclear. The whole sentence should be corrected. Response 3: Okay, we have rewritten this sentence in the revised manuscript. Please refer to lines 299-300 in the updated manuscript for these changes.
Comments 4: Line 316: It is unclear what connective morphology is supposed to mean. It should be named precisely. It should be clearly indicated in Fig. 7D. Response 4: Yes, we acknowledge that the previous descriptions are unclear. The accompanying image set also caused confusion; therefore, we have revised the image set to better illustrate the structure of the dioptric apparatus in the compound eye of female C. gravelyi. Please refer to the updated Figure 7.
Comments 5: Line 365-366: Please indicate it exactly in figures. The data in the text does not correspond to the figures. Response 5: We believe that the question arose due to an omission of figures during the revision process by the corresponding author. To address this, we have added new annotations and a revised set of images to better illustrate our points in the updated manuscript. Please refer to lines 379-381 for these changes.
Comments 6: Line 389: What do the colors green and yellow mean? Response 6: Okay, we acknowledge that the previous descriptions were unclear. To address this, we have revised the description of the retinula cells and provided a clearer explanation. Specifically, in lines 363-364 of the updated manuscript, retinula cells located at the center of the photoreceptive area are now labeled lime-green, while other retinula cells are assigned red and steel-blue for better distinction.
Comments 7: Line 394: It is difficult to understand where the cell is displacing. What does the ommatidial periphery mean? Response 7: In response to Comment 6, we have thoroughly revised the results section to eliminate confusion in the updated manuscript. For your reference, please see lines 341-353 for these changes.
Comments 8: Line 402-411: This is a description of the retinal cells not supported by documentation. In Fig. 8 B, only R8 and R9 are indicated, whereas the other retinal cells mentioned in this paragraph are not shown. Figure 9 A-C also does not show the changes of the R2, R6, etc. Response 8: We acknowledge that the key issue was the omission of an image set documenting retinal cell dynamics from distal to proximal in the original manuscript. To solve this, we have added the missing image set in the revised manuscript. Please refer to new Figure 9 in the updated file for these additions.
Comments 9: Line 417-419, Please number and show the arrangement of these fibers. Response 9: Okay, following your advice, we have reorganized the numbering based on a clockwise orientation. However, it is important to note that this new arrangement does not align with the retinal cell sequence, as we have yet to determine which axons specifically penetrate individual retinal cells.
|
||
|
|
||
|
4. Response to Comments on the Quality of English Language |
||
|
Point 1: The English is fine and does not require any improvement. |
||
|
Response 1: While we did not receive specific feedback on the English quality in the original manuscript, we have improved all language-related issues by engaging a professional language service company in China (https://www.liwenbianji.cn/) based on previous reviewer comments. Thank you for your consideration.
|
||
|
|
||
|
|
||

Round 2
Reviewer 1 Report
Comments and Suggestions for Authors
The authors have done the work of revising the manuscript. I have no additional comments.
Author Response
Comment 1: All figures and tables can be improved.
Response 1: We sincerely thank Reviewer 1 for his/her positive assessment and insightful suggestions. In response, we have carefully enhanced all figures and two tables in the updated manuscript, enhancing the manuscript’s scientific rigor and clarity. We are truly grateful for the reviewer’s expertise, which has been invaluable in refining our work.